# Atorvastatin Inhibits Endothelial PAI-1-Mediated Monocyte Migration and Alleviates Radiation-Induced Enteropathy

**DOI:** 10.3390/ijms22041828

**Published:** 2021-02-12

**Authors:** Seo Young Kwak, Sunhoo Park, Hyewon Kim, Sun-Joo Lee, Won-Suk Jang, Min-Jung Kim, SeungBum Lee, Won Il Jang, Ah Ra Kim, Eun Hye Kim, Sehwan Shim, Hyosun Jang

**Affiliations:** National Radiation Emergency Medical Center, Laboratory of Radiation Exposure and Therapeutics, Korea Institute of Radiological and Medical Science, Seoul 01812, Korea; ksy@kirams.re.kr (S.Y.K.); sunhoo@kirams.re.kr (S.P.); hw0227@kirams.re.kr (H.K.); sjlee@kirams.re.kr (S.-J.L.); wsjang@kirams.re.kr (W.-S.J.); kimmj74@kirams.re.kr (M.-J.K.); sblee@kirams.re.kr (S.L.); zzang11@kirams.re.kr (W.I.J.); ahrakim@kirams.re.kr (A.R.K.); kimeunhye@kirams.re.kr (E.H.K.)

**Keywords:** radiation-induced enteropathy, atorvastatin, plasminogen activator inhibitor-1, endothelial cell, monocyte migration

## Abstract

Intestinal injury is observed in cancer patients after radiotherapy and in individuals exposed to radiation after a nuclear accident. Radiation disrupts normal vascular homeostasis in the gastrointestinal system by inducing endothelial damage and senescence. Despite advances in medical technology, the toxicity of radiation to healthy tissue remains an issue. To address this issue, we investigated the effect of atorvastatin, a commonly prescribed hydroxy-3-methylglutaryl-coenzyme A reductase inhibitor of cholesterol synthesis, on radiation-induced enteropathy and inflammatory responses. We selected atorvastatin based on its pleiotropic anti-fibrotic and anti-inflammatory effects. We found that atorvastatin mitigated radiation-induced endothelial damage by regulating plasminogen activator inhibitor-1 (PAI-1) using human umbilical vein endothelial cells (HUVECs) and mouse model. PAI-1 secreted by HUVECs contributed to endothelial dysfunction and trans-endothelial monocyte migration after radiation exposure. We observed that PAI-1 production and secretion was inhibited by atorvastatin in irradiated HUVECs and radiation-induced enteropathy mouse model. More specifically, atorvastatin inhibited PAI-1 production following radiation through the JNK/c-Jun signaling pathway. Together, our findings suggest that atorvastatin alleviates radiation-induced enteropathy and supports the investigation of atorvastatin as a radio-mitigator in patients receiving radiotherapy.

## 1. Introduction

In therapeutic practice, radiation affects numerous types of normal ‘healthy’ tissues surrounding the targeted tumor tissue. Reducing the toxicity of radiation to normal tissues is critical to improving the therapeutic benefits of radiotherapy and patient quality of life [1,2]. One of the most radiosensitive organs in the body is the intestine. Exposure of the intestine to radiation can cause severe damage to epithelial, endothelial, and neuronal cells, resulting in radiation-induced enteropathy [3,4,5,6]. Individuals with radiation-induced enteropathy frequently develop clinical symptoms, such as diarrhea, vomiting, fatigue, nutritional deficiency, and rectal bleeding [7,8,9]. Although these are serious clinical concerns, there are no FDA-approved drugs for radiation-induced enteropathy. 

The endothelium is a crucial component of acute/chronic radiation-induced gastrointestinal syndrome [10,11,12]. Irradiation alters vascular homeostasis by inducing apoptosis, senescence, defective angiogenesis, abnormal vascular permeability, and acquisition of a pro-inflammatory phenotype [13]. Senescence of endothelial cells results in a senescence-associated secretory phenotype (SASP), characterized by the production of cytokines, proteins, and other factors that cause dysfunction in adjacent cells or lead to inflammation [13].

Plasminogen activator inhibitor-1 (PAI-1) belongs to the serine protease inhibitor family, regulates fibrinolysis, and inhibits tissue plasminogen activator and urokinase [14]. Additionally, PAI-1 is a primary component of SASP and regulates cellular senescence. PAI-1 is mainly produced by the endothelium and is involved in pathophysiological conditions, including inflammation [15], oxidative stress [16], fibrosis [17,18], and macrophage adhesion/migration [19,20]. Several studies have reported radiation-induced upregulation of PAI-1 in endothelial cells in both in vitro and in vivo systems [21,22]. PAI-1 genetic deficiency in mice limits the severity of radiation-induced intestinal injury and improves dermal wound healing after irradiation [21,23]. Moreover, endothelial-specific, but not epithelial-specific, genetic deficiency of PAI-1 limits the severity of radiation enteropathy with endothelial damage in mice [24]. Therefore, endothelial PAI-1 is critically implicated in radiation-induced enteropathy.

Statins are a class of lipid-lowering medications that competitively inhibit cholesterol synthesis via hydroxy-3-methylglutaryl-coenzyme A (HMG-CoA) reductase [25]. Statins have been reported to have therapeutic effects on radiation-induced intestinal damage by inhibiting inflammatory response and fibrotic changes in mouse models [26,27,28,29,30]. Statins also suppress PAI-1 production by regulating NF-κB, MAPK, and PI3K/AKT [31,32,33,34] pathways. Atorvastatin is a statin that is routinely used to treat patients with hypercholesterolemia and atherosclerosis. Functionally, atorvastatin has pleiotropic effects on the fibrinolytic system of vascular smooth muscle and endothelial cells [35] and the anti-inflammatory system [36]. However, atorvastatin’s effects and the molecular mechanisms underlying radiation-induced endothelial damage are not fully understood.

In this study, we investigated the therapeutic effects of atorvastatin on acute radiation-induced enteropathy, focusing on the effects of atorvastatin on endothelial cells. We show that endothelial cells exhibit increased senescence in response to radiation that is attenuated by atorvastatin treatment. Downregulation of the JNK/c-Jun/PAI-1 pathway by atorvastatin results in a diminished inflammatory response and the recovery of epithelial barrier integrity. Importantly, atorvastatin demonstrated therapeutic effects against acute radiation-induced enteropathy in mice. These results suggest that atorvastatin is effective against acute radiation-induced enteropathy, downregulates PAI-1 expression, and reduces trans-endothelial migration of monocytes, thus attenuating inflammation and improving radiation-induced damage.

## 2. Results

### 2.1. Atorvastatin Mitigates Radiation-Induced Endothelial Cell Dysfunction

To investigate the effects of atorvastatin on radiation-induced endothelial damage, we performed 10 Gy irradiation to human umbilical vein endothelial cells (HUVECs) and treated atorvastatin. Previously, we and our colleagues reported that 10 Gy irradiated HUVECs sufficiently emerged radiation-induced damaged endothelial phenotypes [26,37]. We first examined HUVEC tube formation to assess angiogenic capacity. Whereas irradiation decreased the endothelial tube formation ability of HUVECs, atorvastatin treatment markedly improved radiation-induced tube formation defects (Figure 1A). Invasion and migration of endothelial cells is critical for vascular remodeling and angiogenesis. We also investigated whether atorvastatin could influence the invasion capacity in irradiated endothelial cells. Irradiated HUVECs demonstrated reduced invasive ability compared with the control group, however atorvastatin treatment dramatically increased the number of invasive endothelial cells after irradiation (Figure 1B). Senescence is another hallmark of endothelial cell dysfunction and is known to be induced by irradiation in HUVECs. By evaluating β-galactosidase activity, we found that irradiation induced cellular senescence in HUVECs, which was reversed by atorvastatin treatment (Figure 1C). These results suggest that atorvastatin improves radiation-induced endothelial dysfunction, including defective angiogenesis and cellular senescence.

### 2.2. Atorvastatin Inhibits PAI-1 Expression in Irradiated HUVECs and Suppresses Transendothelial Migration of THP-1 Cells

Senescent endothelial cells secrete cytokines and chemokines that affect damaged tissues and contribute to SASP [38,39,40,41]. PAI-1, a prominent component of SASP, is an essential mediator of inflammation in response to intestinal radiation toxicity [21,24,42,43]. To investigate the effect of atorvastatin on radiation-induced PAI-1 expression, we performed Western blot and real-time RT-PCR. We found that atorvastatin significantly reduced radiation-induced PAI-1 mRNA and protein levels in HUVECs (Figure 2A,B). We also evaluated the level of secreted PAI-1 in conditioned media from HUVECs. We found that irradiation also significantly increased secreted PAI-1 protein levels compared to the control group, and atorvastatin treatment markedly decreased PAI-1 secretion in irradiated HUVECs (Figure 2C,D). PAI-1 originating from endothelial cells has been directly and indirectly associated with the inflammation response through monocyte adhesion and migration to the damaged tissue [19,44]. To evaluate a direct effect of PAI-1 in the trans-endothelial migration of monocytes, we used a THP-1 (human monocytic cell line) and HUVEC co-culture system. We observed that HUVECs treated with recombinant PAI-1 recruited a greater number of THP-1 cells (Figure 2E). We wondered whether regulation of PAI-1 by atorvastatin could inhibit monocyte adhesion and migration in irradiated HUVECs. Indeed, irradiation significantly increased THP-1 cell migration compared with non-irradiated HUVECs, and atorvastatin treatment markedly inhibited THP-1 cell migration in irradiated HUVECs (Figure 2F). These results suggest that atorvastatin downregulates PAI-1 expression in irradiated endothelial cells and suppresses trans-endothelial migration of monocytes.

### 2.3. Atorvastatin Suppress PAI-1 Expression Through JNK–c-Jun Signaling in Irradiated Endothelial Cells

PAI-1 expression is regulated by MAPK signaling through p38 and JNK [45,46,47,48,49,50]. Based on our finding that atorvastatin reduces PAI-1 expression, we hypothesized that atorvastatin may inhibit radiation-induced MAPK activation in endothelial cells. Consistent with this hypothesis, atorvastatin treatment effectively inhibited radiation-induced JNK activation in HUVECs but did not affect p38 levels (Figure 3A). In addition, nuclear expression of phosphorylated c-Jun was reduced by atorvastatin in irradiated HUVECs (Figure 3B). Immunofluorescence analysis also showed increased nucleus translocation of phosphorylated c-Jun in the irradiated HUVECs (Figure 3C). In quantification of fluorescence intensity, atorvastatin significantly reduced phosphorylated c-Jun activity in the nucleus of irradiated HUVECs.

To clarify the involvement of the JNK pathway in atorvastatin-regulated PAI-1 expression, we performed experiments using anisomycin, which is a well-known JNK activator. Anisomycin treatment reversed the effect of atorvastatin on radiation-induced PAI-1 production (Figure 4A,B). Furthermore, activation of the JNK pathway in the atorvastatin-treated group did not inhibit nuclear translocation of phosphorylated c-Jun (Figure 4C). In quantification of fluorescence intensity, anisomycin treatment significantly activated nuclear translocation of phosphorylated c-Jun in atorvastatin-treated irradiated HUVECs (Figure 4C). Next, we investigated whether atorvastatin-induced JNK downregulation was involved in PAI-1 expression and monocyte migration. Irradiation of HUVECs significantly increased the number of trans-endothelial migrated THP-1 cells compared with control. Atorvastatin alleviated radiation-induced monocyte migration, but anisomycin treatment abrogated that effect (Figure 4D). Taken together, these findings suggest that atorvastatin inhibits PAI-1 expression by regulating the JNK pathway in irradiated endothelial cells and suppresses trans-endothelial migration of monocytes.

### 2.4. Atorvastatin Alleviates Radiation-Induced Intestinal Damage In Vivo

To investigate the in vivo effect of atorvastatin on radiation-induced enteropathy, we exposed the abdomen of mice to 13.5 Gy of radiation using an X-rad-320. Six days after irradiation, we evaluated the therapeutic effect of atorvastatin by histological examination of villi length and crypt destruction as well as by histological scoring of epithelial structural damage, vascular dilation, and inflammatory cell infiltration in the mucosa and submucosa. Villi length and crypt number were markedly higher in atorvastatin-treated irradiated mice compared to the irradiated group (Figure 5A,D,E). Immunohistochemistry for Ki-67, a proliferation marker, also showed that atorvastatin-treated irradiated mice had elevated numbers of Ki-67 positive cells compared to the irradiated group (Figure 5B). To determine whether atorvastatin affected intestinal barrier function after irradiation, we assessed expression of the tight junction component claudin 3 (CLDN3) and bacterial translocation in mesenteric lymph nodes. Expression of Cldn3 in the intestine was broadly localized in the epithelium of the atorvastatin-treated irradiated group compared to the irradiated group (Figure 5C). Consistent with reduced integrity of the intestinal barrier, the number of bacterial colonies in the mesenteric lymph nodes was significantly increased in the irradiated group compared to the control group. In contrast, atorvastatin treatment decreased bacterial translocation compared with the irradiated group (Figure 5F). These results suggest that atorvastatin alleviates radiation-induced enteropathy and improves intestinal barrier recovery in a mouse model.

### 2.5. Atorvastatin Inhibits PAI-1 Expression and Attenuates Inflammation Response in Radiation-Induced Intestinal Injury

To measure the production of PAI-1 in our mouse model, we performed an enzyme-linked immunosorbent assay (ELISA) using plasma and intestinal tissue from the mice in each group. Plasma and intestinal PAI-1 levels were significantly higher in irradiated mice than in control mice. In contrast, atorvastatin treatment significantly reduced PAI-1 protein levels (Figure 6A). As JNK inactivation by atorvastatin inhibited PAI-1 production in the irradiated HUVEC, we evaluated phospho-JNK expression in the endothelium (CD31) on the intestinal tissue. The activity of JNK on endothelium was increased in irradiated group compared to the control. However, atorvastatin-treated irradiated group showed reduction of JNK activation in endothelium (Figure 6B). Radiation-induced enteropathy is characterized by an inflammatory response with increased inflammatory cell infiltration and cytokine expression [30,51]. Our in vitro results as well as the reports of others have shown that increased PAI-1 enhanced monocyte adhesion and migration, which contributed to inflammatory responses [19,45]. Therefore, we assessed the effect of atorvastatin on radiation-induced inflammatory cell infiltration and the inflammatory response in the intestinal tissue of mice exposed to radiation and atorvastatin. We evaluated myeloperoxidase (MPO) and CD68 as markers of activated neutrophils and macrophages, respectively. We found that MPO and CD68-positive cells increased in response to radiation compared to the control group. In contrast, inflammatory cell accumulation in the irradiated intestine was markedly reduced in atorvastatin-treated irradiated mice (Figure 6C,D). We also determined that expression of the inflammatory cytokines and chemokines metalloprotease 9 (Mmp9), interleukin 1b (Il1b), tumor necrosis factor a (Tnfa), and monocyte chemoattractant protein 1 (Mcp1) was markedly elevated in irradiated intestinal tissue compared with control. The atorvastatin-treated irradiated group showed reduced expression of these inflammatory cytokines and chemokines compared to the irradiated group (Figure 6E). Taken together, our findings demonstrate that atorvastatin alleviates inflammatory responses by regulating PAI-1 production in radiation-induced enteropathy.

## 3. Discussion

Atorvastatin, an FDA-approved small molecule drug, is an HMG-CoA reductase inhibitor that is commonly used to lower cholesterol. Atorvastatin is also reported to have remarkable benefits in the prevention and treatment of cardiovascular disease [52,53]. Radiotherapy is currently a leading therapeutic approach for pelvic cancers. However, abdominal irradiation causes toxicity in the surrounding healthy gastrointestinal tissue with short and long term side effects. Here, we demonstrate that atorvastatin recovered radiation-induced endothelial dysfunction by promoting angiogenic and anti-senescent activity and inhibited PAI-1 expression in endothelial cells. Moreover, we show that atorvastatin mitigated radiation-induced enteropathy by attenuating epithelial destruction and immune cell infiltration in mice.

The vascular endothelium is a principal checkpoint for radiation-induced pathogenic processes in both normal and tumor tissue and can thus be evaluated to determine therapy efficiency [11,54,55,56]. Radiation directly affects the vasculature by causing endothelial cell apoptosis and senescence. Senescent endothelial cells secrete cytokines, proteins, and other factors that cause dysfunction in adjacent cells or lead to a chronic inflammatory state. In the present study, we irradiated endothelial cells to investigate the cell specific effect of atorvastatin after radiation damage. We found that atorvastatin mitigated radiation-induced endothelial damage by activating angiogenesis and reducing senescence in endothelial cells. Our results demonstrate that atorvastatin can effectively protect against endothelial damage following radiation. Therefore, we explored the mechanisms of the radio-mitigative effect of atorvastatin in endothelial cells.

Our results, along with several reports [21,53], indicate that PAI-1 is a critical component of the SASP in irradiated endothelial cells. In our study, PAI-1 expression and secretion were effectively downregulated by atorvastatin in irradiated HUVECs. In addition, we showed that PAI-1 expression decreased in intestinal tissues and plasma of atorvastatin-treated irradiated mice, providing in vivo confirmation that atorvastatin impairs the production and secretion of PAI-1.

The MAPK pathway is involved in the production of PAI-1 [46,47,48,49,50,57,58]. In irradiated endothelial cells, we observed activation of the JNK pathway, but not p38 signaling, and induction of PAI-1. We also demonstrated activation of the JNK pathway by nuclear translocation of phosphorylated-c-Jun in irradiated endothelial cells. Radiation-induced JNK activation, c-Jun nuclear translocation, and PAI-1 expression were inhibited by atorvastatin treatment in irradiated endothelial cells. In contrast, pharmacological activation of JNK reversed the inhibitory effect of atorvastatin on PAI-1 expression in irradiated endothelial cells. We suggest that atorvastatin inhibits PAI-1 production by downregulating the JNK-c-Jun pathway after irradiation.

Mice deficient for endothelial PAI-1 exhibit limited severity of radiation-induced intestinal injury with endothelial damage [24]. Although the role of endothelial PAI-1 in radiation-induced inflammation has not been established, increased PAI-1 expression has been related to inflammatory lesions [19,59,60,61]. In this study, we demonstrate that radiation-induced PAI-1 directly regulates trans-endothelial migration of monocytes. Interestingly, monocyte migration was inhibited by atorvastatin treatment in irradiated endothelial cells, and this effect was abolished by anisomysin-mediated JNK activation. Furthermore, atorvastatin inhibited radiation-induced neutrophil and macrophage migration to the intestinal tissue, and downregulation of inflammatory cytokines and chemokines could be detected in the atorvastatin-treated irradiated intestine. Taken together, our results imply that atorvastatin regulates PAI-1 production and attenuates radiation-induced inflammatory response in the irradiated intestine. 

In this study, we provide evidence that supports the use of atorvastatin as a promising radio-mitigator of endothelial damage and enteropathy. We further provide key insight into the mechanism of atorvastatin action by demonstrating that atorvastatin inhibits PAI-1 expression through the JNK pathway in irradiated endothelial cells. Although many strategies have been developed to address acute radiation-induced gastrointestinal damage, none have gained FDA approval for human applications. Thus, there is a critical need to repurpose FDA-approved drugs, such as atorvastatin, as novel countermeasures to radiation-induced gastrointestinal injury. Our study supports the use of atorvastatin as a potential novel countermeasure to radiation-induced enteropathy.

## 4. Materials and Methods

### 4.1. Cell Culture and Reagents

HUVECs (Lonza, Basel, Switzerland) were maintained in EGM-2 medium supplemented with endothelial growth kit components (Lonza). Human monocytic cells (THP-1) were maintained in RPMI-1640 (Lonza) supplemented with 10% fetal bovine serum (FBS; Gibco, USA) and 10% antibiotics (Thermo Fisher Scientific, Waltham, MA, USA). All cells were grown in a humidified incubator at 37 °C with 5% CO_2_. An antibody specific for PAI-1 was purchased from BD Biosciences (San Jose, CA, USA). Antibodies against phospho-JNK, phospho-c-Jun, JNK, c-Jun, phospho-p38, and p38 were purchased from Cell Signaling Technology (Danvers, MA, USA). Beta-actin and histone H3 antibodies were purchased from Santa Cruz Biotechnology. Atorvastatin and anisomycin (MAPK activator) were obtained from Sigma-Aldrich (St Louis, MO, USA). Recombinant human PAI-1 was purchased from Peprotech (Rocky Hill, NJ, USA). 

### 4.2. Irradiation and Treatment

HUVECs were seeded in 35 mm dishes and maintained for 24 h. Next day, the cells were exposed to 10 Gy of radiation using a ^137^Cs γ-ray source (Atomic Energy of Canada, Ltd.,Ontario, Canada) with a dose rate of 3.81 Gy/min (Gammacell 3000 Elan, MDS Nordion, Ottawa, Canada). Atorvastatin was immediately treated with the irradiated HUVECs. All in vitro experiments were performed after 24 h treated with irradiation and atorvastatin.

To obtain conditioned media, irradiated HUVECs were treated with atorvastatin and incubated for 24 h. After incubation, media was exchanged with fresh EBM-2 media.

### 4.3. Tube Formation Assay

Matrigel (BD Biosciences) was added to 24-well plates and allowed to solidify at 37 °C. Once Matrigel solidified, prepared HUVECs were re-seeded into Matrigel-coated wells and treated with or without atorvastatin for 6 h. Angiogenic ability including the total segment length (i.e., sum of the length of the section delimited by one junction and one extremity) and number of the master junction (i.e., total number of the junctions that is not only implicated in a branch) was observed under a light microscope and plotted using Image J software (National Institutes of Health).

### 4.4. Invasion Assay

Irradiated HUVECs were re-seeded 24 h later into Matrigel-coated trans-wells (Corning, NY, USA) and then treated with or without atorvastatin. The cells were allowed to invade through Matrigel for 24 h. After invasion, the filters were stained with Diff-Quik (Marz-Dade, Düdingen, Switzerland). The invading cells in nine different fields were counted under a light microscope. The invasive capacity of HUVECs was normalized to that of the control group.

### 4.5. Senescence-Associated β-Galactosidase Assay (SA-β-gal Assay)

Prepared cells were fixed with 4% paraformaldehyde and subsequently stained with β-galactosidase kit (Cell Signaling Technology) according to the manufacturer’s instructions. 

### 4.6. RNA Extraction and Real-Time RT-PCR

Total RNA was extracted from HUVECs using Tri-reagent (MRC, Cincinnati, OH, USA) according to the manufacturer’s protocol. To detect the expression of genes in intestinal tissue, one piece of flash frozen ileum tissue was subjected to RNA extraction using Tri-reagent. Two micrograms of total RNA were synthesized into cDNA using the AccuPower RT premix (Bioneer, Daejeon, Korea). Synthesized cDNA was amplified using a LightCycler 480 system (Roche, Basel, Switzerland) with specific primers according to the manufacturer’s instructions. Expression levels of each gene were determined using the ddCt method, and GAPDH was used as a housekeeping gene. The sequences of the primers were as follows: human PAI-1, 5′-CCCAGCTCATCAGCCACT-3′ (forward) and 5′-GAGGTCGACTTCAGTCTCCAG-3′ (reverse); mouse Mmp9, 5′-GCCCTGGAACTCACACGACA-3′ (forward) and 5′-TTGGAAACTCACACGCCAGAAG-3′ (reverse); mouse Il1β, 5′-GCAACTGTTCCTGAACTCA-3′ (forward) and 5′-CTCGGAGCCTGTAGTGCAG-3′ (reverse); mouse Tnfa, 5′-GCCTCTTCTCATTCCTGCTT-3′ (forward) and 5′-CACTTGGTGGTTTGCTACGA-3′ (reverse); mouse Mcp1, 5′-AGGTCCCTGTCATGCTTCT-3′ (forward) and 5′-CTGCTGGTGATCCTCTTGT-3′ (reverse), human GAPDH, 5′-GGACTCATGACCACAGTCCATGCC-3′ (forward) and 5′-TCAGGGATGACCTTGCCCACAG-3′ (reverse); and mouse β-actin, 5′-TCCCTGGAGAAGAGCTATGA-3′ (forward) and 5′-CGATAAAGGAAGGCTGGAA-3′ (reverse).

### 4.7. Western Blot Analysis

Prepared HUVECs were homogenized with RIPA buffer (Thermo Fisher Scientific) and resolved by SDS-PAGE. To detect secreted PAI-1, conditioned media was collected, mixed with 5× sample buffer, and boiled for Western blot analysis. The protein content was normalized by Bradford assay. Prepared cells were resolved by SDS-PAGE, and transferred to PVDF membranes (Millipore, USA). After transfer, the membranes were blocked with 5% skim milk in TBS-T (10 mM Tris/HCl, pH 8.0, 150 mM NaCl, 0.05% Tween-20) for 1 h, and incubated with the specific primary antibody in the blocking solution. Western blotting was performed with specific antibodies for PAI-1 (BD bioscience), phospho-JNK, JNK, phospho-p38, p38 (Cell signaling technology), and β-actin (Santa Cruz Biotechnology). The membranes were incubated with horse-radish peroxidase-conjugated secondary antibody in TBS-T followed by chemiluminescence detection using an enhanced chemiluminescence (ECL) system (Thermo Fisher Scientific, USA).

### 4.8. Enzyme-Linked Immunosorbent Assay (ELISA)

To quantify PAI-1, conditioned media was collected and briefly centrifuged at 1300× *g* to remove cell debris. The supernatant was subjected to ELISA (R&D Systems, Minneapolis, MN, USA) according to the manufacturer’s instructions. For in vivo experiments, mouse plasma was used to quantify secreted PAI-1, and small intestine tissues were used to quantify local PAI-1. Small intestine tissues were homogenized with RIPA buffer and then subjected to ELISA to quantify cellular PAI-1 according to the manufacturer’s instructions (R&D Systems). The samples were normalized by Bradford assay (Thermo Fisher Scientific).

### 4.9. Trans-Endothelial Migration of Monocytes

THP-1 trans-endothelial migration assay was performed using Matrigel-coated trans-well plates (Corning, USA). To evaluate trans-endothelial migration of THP-1, the irradiated or control HUVECs (5 × 10^4^) were seeded onto Matrigel-coated trans-well and treated with atorvastatin or anisomycin. Serum-starved THP-1 cells were stained with carbosyfluorescein diacetate succinimidyl ester (Thermo Fisher Scientific) and subsequently added to the trans-well for co-culture with the prepared HUVECs (4:1 ratio). After 4 h, trans-wells were washed and fixed in 4% paraformaldehyde and mounted with Vectashield Mounting Medium (Vector Laboratories, Burlingame, CA, USA). Fluorescence images showing the trans-endothelial migration of THP-1 cells (green) were observed under the microscope (Leica, Germany). The green-positive cells in five random microscopic fields of the group were quantified and plotted as a bar graph.

### 4.10. Preparation of Nuclear and Cytosolic Fractions

Nuclear fractions from HUVECs were prepared using the Nuclear and Cytoplasmic Extraction Reagent (NE-PER; Thermo Fisher Scientific) kit according to the manufacturer’s instructions. Western blotting was performed with specific antibodies for phospho-c-Jun (Cell Signaling Technology) and histone H3 (Santa Cruz Biotechnology).

### 4.11. Immunofluorescence

HUVECs were seeded on cover slips and exposed to radiation prior to treatment with or without atorvastatin. After 24 h, prepared cells were fixed with 4% paraformaldehyde and permeabilized (0.1% triton X-100, 5% BSA in PBS) for 1 h at room temperature. The cells were incubated with the specific antibody for phospho-c-Jun at 4 °C. After a brief wash, the cells were incubated with Alexa Fluor 488-labeled secondary antibody (Life Technologies, Carlsbad, CA, USA) for 1 h at room temperature. Finally, the samples were washed with PBS and mounted with Vectashield Hard Set mounting medium with DAPI (4′6-diamidino-2-phenylindole; Vector Laboratories). The fluorescence intensities of phospho-c-Jun in the nucleus were analyzed using image J and plotted as a bar graph.

### 4.12. In Vivo Experiment and Irradiation

Specific pathogen-free (SPF) male C57BL/6 mice (7-week-old) were obtained from Harlan Laboratories (Indianapolis, IN, USA) and housed under SPF conditions at an animal facility of the Korea Institute of Radiological and Medical Sciences (KIRAMS). All mice were maintained in a temperature-controlled room with a 12-h light/dark cycle to acclimate. Mice were assigned to the following groups: control (Con), irradiated (IR), and IR with atorvastatin treatment (IR + Atorva). All animal experiments were approved and performed in accordance with the guidelines of the Institutional Animal Care and Use Committee of KIRAMS. Animals were anesthetized with 85 mg/kg alfaxalone (Alfaxan^®^, Careside, Gyeonggi-do, Korea) and 10 mg/kg xylazine (Rompun^®^ Bayer Korea, Seoul, Korea. Mice were irradiated in the abdomen with single 13.5 Gy dose using an X-RAD 320 X-ray irradiator (Softex, Gyeonggi-do, Korea). Dosage and rate of delivery were strictly monitored (UNIDOS E Universal Dosemeter; PTW-Freiburg, Freibug, Germany). Atorvastatin (10 mg/kg/day; Pfizer Inc., USA) was given orally for 6 days immediately after irradiation.

### 4.13. Histological Analysis

The ileum tissue was fixed with a 10% neutral buffered formalin solution, embedded in paraffin wax, and sectioned at a thickness of 4 µm. The sections were stained with hematoxylin and eosin, and evidence of intestinal mucosal injury was quantified (0 = none, 1 = mild, 2 = moderate, 3 = high). The histological severity of radiation-induced intestinal damage was assessed by the degree of epithelial architecture maintenance, crypt damage, vascular dilation, and infiltration of inflammatory cells in the lamina propria. For immunohistochemical analysis, unstained sections were subjected to heat-induced antigen retrieval and treated with 0.3% hydrogen peroxide in methyl alcohol to block endogenous peroxidase activity. After blocking with 5% normal goat serum (Vector ABC Elite kit; Vector Laboratories), the sections were incubated with specific antibodies for anti-Ki-67 (Acris), anti-Claudin 3 (Invitrogen, Carlsbad, CA, USA), anti-CD68 (Abcam), and anti-myeloperoxidase (MPO; Abcam). After three washes in PBS, the sections were incubated with a horseradish peroxidase-conjugated secondary antibody (Dako, Carpinteria, CA, USA). The peroxidase reaction was developed using a diaminobenzidine substrate (Dako) according to the manufacturer’s instructions. Hematoxylin was used as a counterstain. 

For fluorescent staining, slides were subjected to antigen retrieval and blocked with 5% goat serum containing 0.1% Triton X-100 for 30 min. The slides were incubated with specific antibodies for phospho-JNK (Cell Signaling Technology) and CD31 (R&D systems) diluted at 1:100 in PBS overnight at 4 °C. After washing with PBS, slides were incubated with Alexa Fluor 488 or 555-labeled secondary antibody (Life Technologies) at room temperature. Finally, the samples were washed with PBS and then mounted with Vectashield HardSet mounting medium with DAPI (Vector Laboratories).

### 4.14. Bacterial Translocation Assay

Mice were sacrificed, and the mesenteric lymph nodes were harvested under sterile conditions. The mesenteric lymph nodes were homogenized with sterile PBS and spread onto MacConkey agar (BD Biosciences) to evaluate the translocation of bacteria from the intestinal lumen. The colonies were counted on all plates.

### 4.15. Statistical Analysis

All data in this study are expressed as the mean ± standard deviation (SD). Statistical analyses were performed using one-way analysis of variance (ANOVA) with Tukey’s multiple comparison test. *p* values < 0.05 were considered statistically significant.

## Figures and Tables

**Figure 1 ijms-22-01828-f001:**
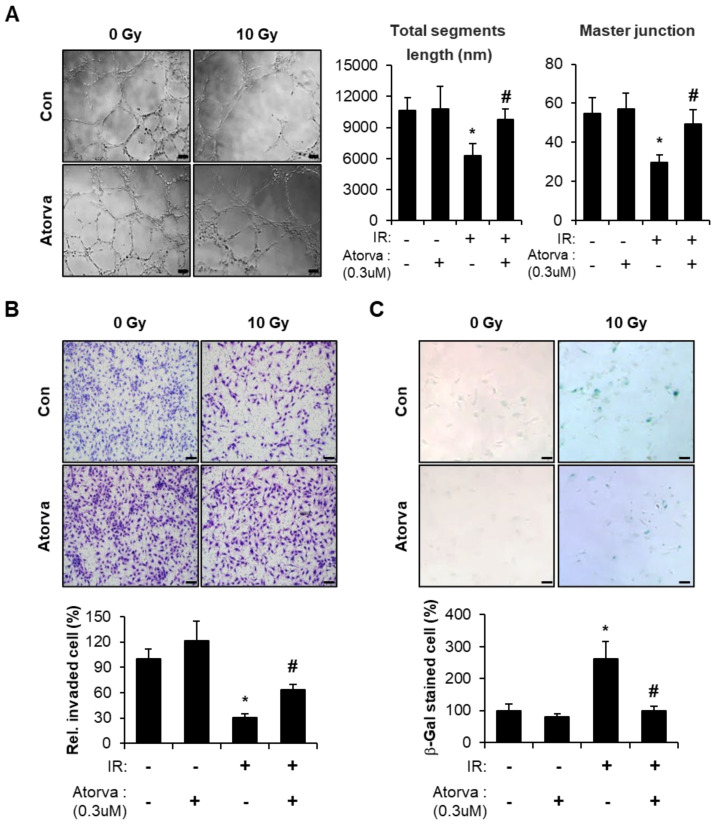
Atorvastatin mitigates radiation-induced endothelial cell dysfunction. (**A**) Tube formation in atorvastatin-treated irradiated human umbilical vein endothelial cells (HUVECs). Representative images of HUVECs seeded on Matrigel in the presence (Atorva) or absence of atorvastatin (left). Total segment length (middle) and number of master junctions (right) were quantified and plotted. (**B**) Invasion in atorvastatin-treated irradiated HUVECs. Representative images of migrated HUVECs on trans-wells (upper). Cells that invaded were quantified and plotted (lower). The invaded cells in nine randomly selected fields were counted under a light microscope. Bars represent the percentage of invaded cells normalized to that of the corresponding control. (**C**) Senescence-associated (SA) β-galactosidase activity in atorvastatin-treated irradiated HUVECs. Representative images were obtained by light microscopy. Senescent cells were quantified and plotted (lower). Data are presented as the mean of ± standard deviation of triplicate experiments. Images are representative of 3 independent experiments. * *p* < 0.05 vs. negative control (Con), ^#^
*p* < 0.05 vs. irradiated control (IR). Scale bars represent 100 μm.

**Figure 2 ijms-22-01828-f002:**
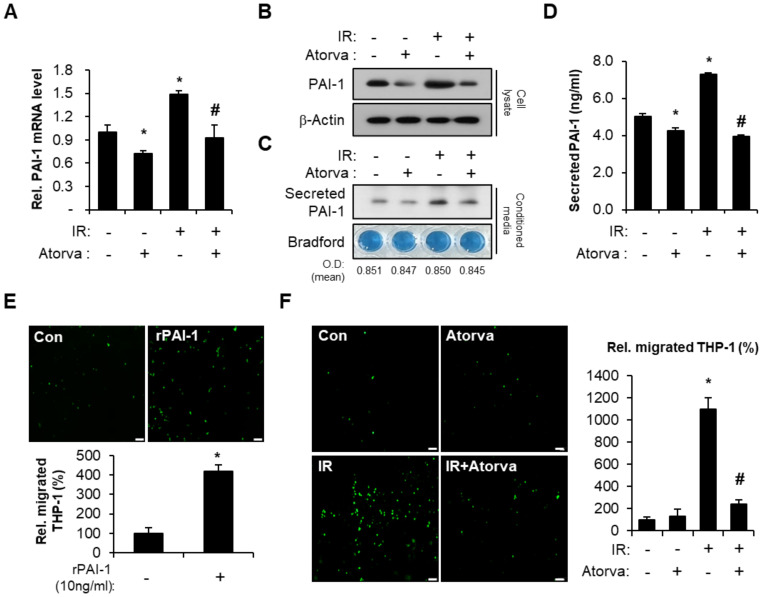
Atorvastatin inhibits PAI-1 production in irradiated endothelial cells and regulates trans-endothelial migration of monocytes. (**A**) mRNA levels and (**B**) protein expression of PAI-1 in control (Con), atorvastatin (Atorva)-treated, irradiated (IR), and IR + Atorva human umbilical vein endothelial cells (HUVECs). (**C**) Western blot analysis and (**D**) Enzyme-linked immunosorbent assay (ELISA) of secreted PAI-1 in conditioned media. The protein content from conditioned media was normalized by Bradford assay. (**E**) Trans-endothelial migration of monocytes in Con and recombinant PAI-1 (rPAI-1)-treated HUVECs. Representative images of migrated CFSE-positive THP-1 cells. The migrated THP-1 cells were counted and plotted. (**F**) Trans-endothelial migration of carbosyfluorescein diacetate succinimidyl ester (CFSE)-stained THP-1 cells co-cultured with Con, Atorva, IR, and IR + Atorva HUVECs. Data are presented as the mean of ± standard deviation of triplicate experiments. Images are representative of 3 independent experiments. * *p* < 0.05 vs. negative control (Con), ^#^
*p* < 0.05 vs. irradiated control (IR). Scale bars represent 100 μm.

**Figure 3 ijms-22-01828-f003:**
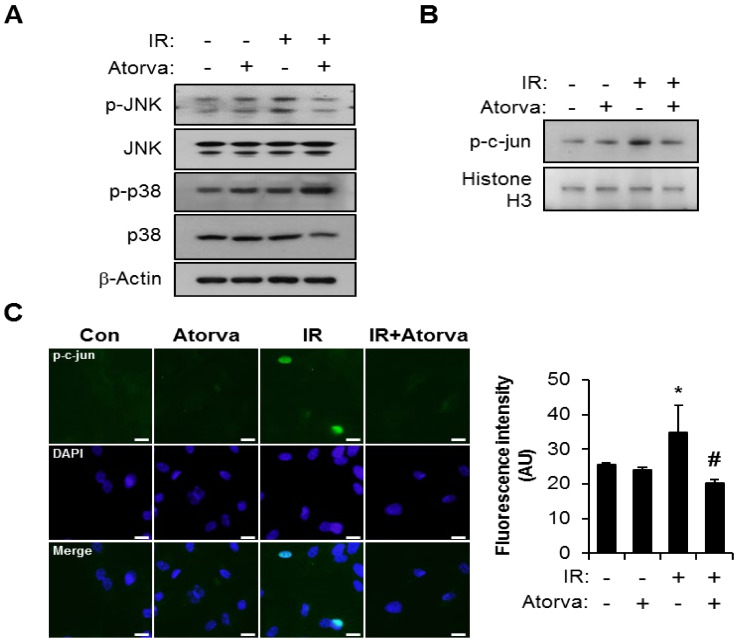
Atorvastatin inhibits radiation-induced JNK/c-Jun signaling. (**A**) Western blot analysis to determine JNK and p38 activation in control (Con), atorvastatin (Atorva)-treated, irradiated (IR), and IR + Atorva HUVECs. (**B**) Western blot analysis to confirm nuclear translocation of phosphorylated c-Jun. Cytoplasmic and nuclear extracts were separated, and western blot analysis was performed to evaluate translocation of phosphorylated c-Jun. (**C**) Immunofluorescence images and the fluorescence intensity of phosphorylated c-Jun. Representative images showing phosphorylated c-Jun (green) and DAPI (blue). Scale bars represent 100 μm. Data are presented as the mean± standard deviation of triplicate experiments. Images are representative of 3 independent experiments. * *p* < 0.05 vs. negative control (Con), ^#^
*p* < 0.5 vs. irradiated control (IR).

**Figure 4 ijms-22-01828-f004:**
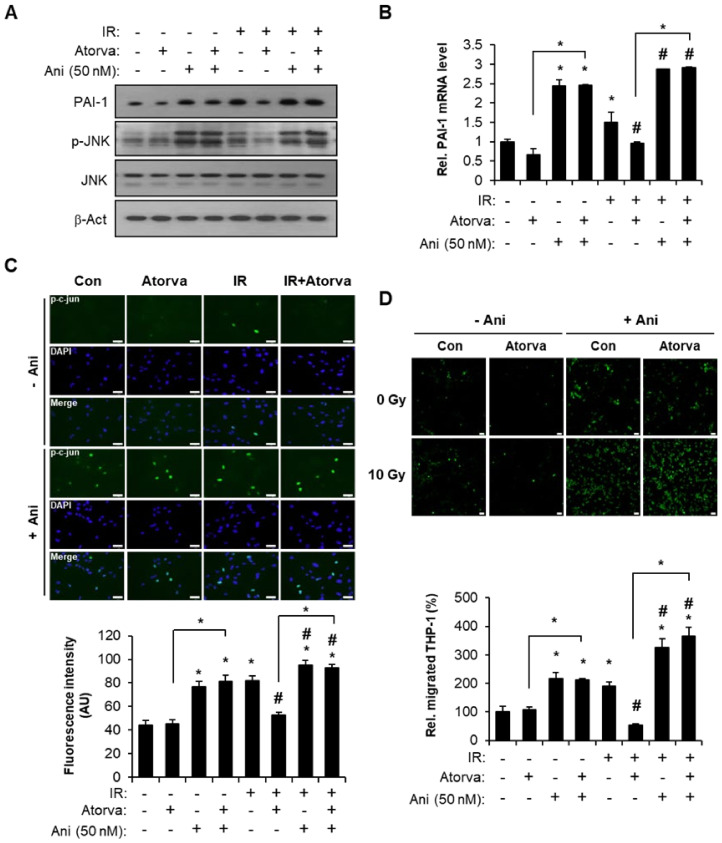
Inactivation of the JNK pathway by atorvastatin regulates PAI-1 production and monocyte migration. (**A**) Western blot analysis of PAI-1 expression and JNK activation in control (Con), atorvastatin-treated (Atorva), irradiated (IR), and IR + Atorva HUVECs in the presence or absence of anisomycin (Ani, 50 nM). (**B**) mRNA levels of PAI-1 and (**C**) immunofluorescence images and the fluorescence intensity of phosphorylated c-Jun (green) in Con, Atorva, IR, and IR + Atorva HUVECs in the presence or absence of Ani. (**D**) Trans-endothelial migration of THP-1 cells in atorvastatin-treated irradiated HUVECs. Representative images of migrated carbosyfluorescein diacetate succinimidyl ester (CFSE)-stained THP-1 cells (green). The number of migrated THP-1 cells was quantified and plotted (lower). Data are presented as the mean of ± standard deviation of triplicate experiments. Images are representative of 3 independent experiments. * *p* < 0.05 vs. negative control (Con), ^#^
*p* < 0.5 vs. irradiated control (IR). Scale bars represent 100 μm.

**Figure 5 ijms-22-01828-f005:**
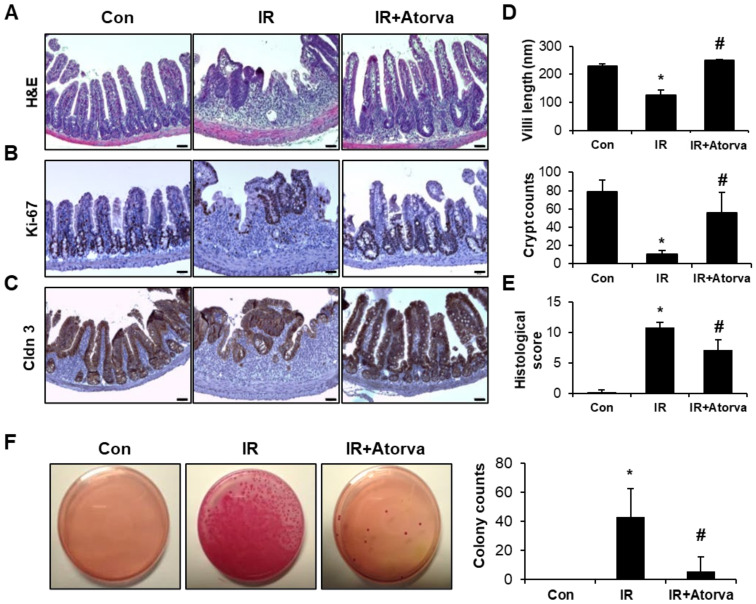
Atorvastatin mitigates radiation-induced intestinal damage in a mouse model. (**A**) Hematoxylin and eosin (H&E) staining; immunostaining for (**B**) Ki-67 and (**C**) Claudin 3 (Cldn3); and (**D**) quantification of villi length and crypts in ileum tissues harvested from control (Con), irradiated (IR), and atorvastatin (Atorva)-treated IR (IR + Atorva) mice 6 days after the local application of 13.5 Gy of irradiation. Scale bars represent 50 μm. (**E**) Histological scoring was assigned based on the degree of epithelial architecture maintenance, crypt damage, vascular enlargement, and infiltration of inflammatory cells in the lamina propria (0 = none, 1 = mild, 2 = moderate, 3 = high) of the ileum of Con, IR, and IR + Atorva groups. (**F**) The number of bacterial colonies from mesenteric lymph node tissue of Con, IR, and IR + Atorva groups. Data are presented as the mean ± standard deviation of the mean; ***n*** = 5 mice per group. * *p* < 0.05 compared to the Con group; ^#^
*p* < 0.05 compared to the IR group.

**Figure 6 ijms-22-01828-f006:**
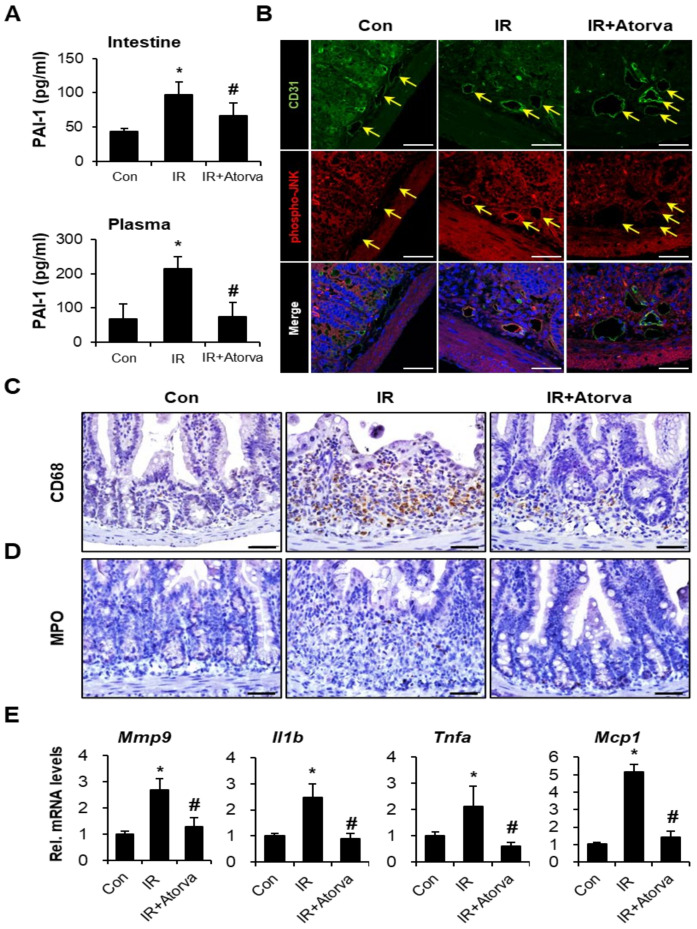
Atorvastatin mitigates the radiation-induced inflammatory response in vivo by inhibiting PAI-1 expression. (**A**) Small intestine and plasma PAI-1 levels in control (Con), irradiated (IR), and atorvastatin (Atorva)-treated IR (IR + Atorva) mice. Immunohistochemistry of (**B**) CD31 (an endothelial cell marker; Green) and phospho-JNK (Red), (**C**) CD68, a monocyte maker, and (**D**) myeloperoxidase (MPO), a neutrophil marker, in intestinal tissue. The yellow arrows indicate CD31-positive endothelial cells. (**E**) mRNA levels of (pro)-inflammatory cytokines Mmp9, Il1b, Tnfa, and Mcp1 in the intestinal tissue of Con, IR, and IR + Atorva groups. Data are presented as the mean ± standard deviation of the mean; ***n*** = 5 mice per group. * *p* < 0.05 compared to the Con group; ^#^
*p* < 0.05 compared to the IR group. Scale bars represent 50 μm.

## Data Availability

The data that support the findings of this study are available from the corresponding author upon reasonable request.

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
