# Peer review of "Atorvastatin Inhibits Endothelial PAI-1-Mediated Monocyte Migration and Alleviates Radiation-Induced Enteropathy"

_ijms, 2021, doi:10.3390/ijms22041828_

Round 1

Reviewer 1 Report

I thank the authors for addressing the comments appropriately.

Author Response

Thank you very much.

Reviewer 2 Report

The authors responded most of the raised questions and improved the manuscript in certain aspects. However, they did not adequately respond to a few points that are important. For instance:

MAJOR POINT NUMBER 4. In the section 4.10., the authors say that “the green-positive cells per five fields were quantified and plotted”. However, no details regarding the microscope objective (e.g., magnification) were given and the readers don’t have an idea how representative five fields are in this context. Please provide more details regarding the cell counting/ microscopy including the number of cells counted per total amount of cells plated.

AUTHORS’ RESPONSE: Line 134; Section 4.9. Responding to the comment, we provided scale bar and the additional information about cell counting and microscopy in the section 4.9.

ISSUE: The authors did not provide any details requested regarding the total number of cells counted versus the number of green cells obtained. This should be feasible since they have nuclei-stained images as well. In addition, details regarding the microscopy (i.e., objective magnification) were not provided in section 4.9 as claimed by the authors.

OTHER ISSUES: In some places of the manuscript in which new modifications were added there are a few English grammar mistakes and typos that need to be corrected.

Author Response

Dr. Luiza Barham

Assistant Editor

International Journal of Molecular Science

February 08, 2021

RE: Ms#IJMS-1061934_revision 2

Dear Dr.Luiza Barham

   Thank you very much for response of February 7, 2021, with regard to our manuscript entitle “Atorvastatin inhibits endothelial PAI-1-mediated monocyte migration and alleviates radiation-induced enteropathy” together with the comments from the reviewers. According to your suggestions, we conducted additional experiments and revised our manuscript.

Our alterations after the reviewers’ comments as follows:

#Reviewer 2

The authors responded most of the raised questions and improved the manuscript in certain aspects. However, they did not adequately respond to a few points that are important. For instance:

ISSUE: The authors did not provide any details requested regarding the total number of cells counted versus the number of green cells obtained. This should be feasible since they have nuclei-stained images as well. In addition, details regarding the microscopy (i.e., objective magnification) were not provided in section 4.9 as claimed by the authors.

  • Thank you for your comments. Line 134; Line 174-175. Responding to the comment, we provided scale bar. We did not perform the total cell counting about transwell migrated cells including HUVECs and THP-1 cell (green-positive cell). In Figure 1B, we identified that atorvastatin treatment increased invasion activity in irradiated HUVECs. And migration activity of THP-1 (green positive cells) decreased in atorvastatin-treated irradiated endothelial cells. According to the data, we estimated more dramatical fold change in the total number of migrated cells versus the number of green cells (THP-1) in IR group and atorvastatin-treated IR group. In this study, we only analyzed trans-endothelial migrated THP-1 cells counting using CFSE stain in the microscopic field. This analysis method has been applied in many papers (Kim et al., 2017. Nutrition Research and Practice; Jin et al., 2014. Biochimica et Biophysica Acta; Monfoulet et al., 2017. Free Radical Biology and Medicine).

OTHER ISSUES: In some places of the manuscript in which new modifications were added there are a few English grammar mistakes and typos that need to be corrected.

  • Responding to the comment, we revised sentences to correct English grammar.

We are grateful that the manuscript has been improved satisfactory and hope that it would be accepted for publication in the International Journal of Molecular Science.

Very sincerely yours,

Hyosun Jang, DVM, PhD,

Senior research

Korea Institute of Radiological and Medical Science 

This manuscript is a resubmission of an earlier submission. The following is a list of the peer review reports and author responses from that submission.

Round 1

Reviewer 1 Report

The authors have addressed an important issue of vascular damage seen in radiation therapy. Although I believe that a lot of work has been put into this paper, it seems to not have been well linked with regards to the mechanisms claimed. My comments are below,

Minor

  1. The abstract does not mention what type of cell lines were used to test the hypothesis. Would suggest relating specific results to either cell culture or animal model.
  2. Introduction section – “Plasminogen activator inhibitor-1 (PAI-1) belongs to the serine protease inhibitor family, regulates fibrinolysis, and inhibits tissue plasminogen activator and urokinase.” Add reference.
  3. The introduction is too long, needs to be presented succinctly in relation to the discussed topic.
  4. Would suggest rearranging all methods related to cell culture first, then followed by all animal experiments.

Major

  1. How were the animal models confirmed to have radiation-induced enteropathy? Any specific markers studied in this regard? If this is an established and validated method to develop enteropathy, please provide a reference.
  2. If the authors thought HUVECs were more translatable to humans, why were animals used? Animal samples seem to be used only for histology. Why these samples were not used to test the expression levels of all the cytokines? How would it compare to the HUVECs results?
  3. Only PAI and anti-inflammatory markers were studied with animal samples. C-Jun/JNK were studied in HUVECs. However, the authors claim that atorvastatin inhibited PAI-1 production following radiation through the JNK/c-Jun signaling pathway. This mechanistic connection claim by testing in different in vivo and in vitro methods may not be acceptable.

Reviewer 2 Report

This manuscript describes a study of the effects of atorvastatin on endothelial PAI-1 in vitro and on radiation enteropathy in a mouse model in vivo. The manuscript addresses an important clinical problem, and statins have potential to reduce radiation injuries. However, some clarifications are required in the manuscript:

  1. In the Introduction, please highlight some of the prior publications that have shown how statins affect PAI-1 expression.
  2. In the Introduction, please highlight some of the prior publications that have shown beneficial effects of statins in radiation enteropathy.
  3. In Figure 1B, please explain how the relative percentages of invaded cells were calculated.
  4. In section 2.2, please spell out (and explain) THP-1 cells.
  5. The legend of Figure 3 describes symbols (* and #) for p-values, but no data are presented in this figure. Were graphs omitted by accident?
  6. In the Discussion, fifth paragraph, it is stated that atorvastatin attenuates radiation-induced inflammatory cell migration by regulating PAI-1 production. However, the authors cannot exclude that other mechanisms have contributed to the effects of atorvastatin in the mouse model. The effects of atorvastatin were probably only in part via regulating PAI-1. Please reword this statement in the manuscript.
  7. Some clarifications are required in the Materials and Methods section:
    1. Section 4.3: Please describe the methods that were used for radiation dosimetry (both of the cesium source for the cell irradiation and of the X-ray source for the animal irradiation).
    2. Section 4.3: Atorvastatin was administered orally for 6 days after irradiation. Please specify when the first dose was administered (immediately after irradiation? 24 hours after irradiation?). Please specify whether atorvastatin was administered once a day.
    3. Section 4.8: Please describe the methods that were used for visualization of bound antibody in the western blots.
    4. Section 4.13: Please describe which part of the small intestine was examined (duodenum, jejunum, or ileum).

Reviewer 3 Report

The manuscript by Kwak et al. regards the capability of atorvastatin to alleviate radiation-induced enteropathy. In particular, the authors tested the ability of atorvastatin to alleviate radiation-induced endothelial cell dysfunction in vitro and in vivo by inhibiting PAI-1 expression. They conclude that this drug mitigates radiation-induced endothelial damage by regulating PAI-1 production and secretion through the JNK/c-Jun signaling pathway and suggest the use of this inhibitor of cholesterol synthesis as a radio-mitigator in patients receiving radiotherapy.

This kind of studies is important for its potential therapeutic implications. The data are clearly written and described. The figures are very well organized and presented, even if they need some more details, corrections and statistical analysis. The manuscript is clearly written and it reads very well. Overall, the shown data support the conclusions of the authors. However, some issues need to be addressed.

In the Results section, for a better reading of the manuscript, the authors should specify in the text (paragraph 2.1) the unit of ionizing radiation utilized and the reason of such choice. Moreover, in paragraph 2.1, lines 8-9, the authors should indicate in the text the kind of assay performed to obtain the data showed in Figure 1B.

In the legend of Figure 2, I think the authors have confused and inverted the text of Figure 2E with 2F.  The text must be checked and corrected.

Then, in the Results section, paragraph 2.4 line 7 the authors report “Villi length and crypt number were markedly higher in atorvastatin-treated irradiated mice compared to the irradiated group (Figure 5A, 5C and 5D)” and then in line 14 “Expression of CLDN3……(Figure 5E)”. In Figure 5C are not considered villi length and crypt number and expression of CLDN3 is shown in Figure 5C. Check and eventually correct the numbers of Figures.

When in the manuscript the authors show western blot data, they should specify how many experiments were performed and show data analysis also in a graph with standard deviation of n experiments to support their conclusions. This is of course particularly important when in western blot data the phosphorylation status is analyzed. In example, the authors in Figure 3B report that “Data are presented as the mean ± standard error of triplicate experiments, but they don’t show any graph”.

Then, also in Figure 3C, the authors should specify how many experiments were performed.

When describing the variability of measurements in a sample, the standard deviation (SD) is the parameter of choice, since SD estimates the variability in the total population and it is independent from the sample size remaining similar when the sample size increases. On the other side, the standard error (SE) describes the uncertainty of the sample mean, giving the precision of the sample mean. Therefore, the data showed in the manuscript should be analyzed considering SD instead of SE.

Reviewer 4 Report

In the present manuscript entitled “Atorvastatin inhibits endothelial PAI-1-mediated monocyte migration and alleviates radiation-induced enteropathy”, Kwak and collaborators show that atorvastatin inhibits PAI-1 production through the JNK/c-Jun signaling pathway following radiation and they suggest that atorvastatin can alleviate radiation-induced enteropathy. Even though their in vitro and in vivo findings are interesting, there are some issues that need to be addressed before I can recommend this paper to be accepted for publication. These issues are grouped into minor and major as shown below:

MINOR:

  1. Please provide scale bars to the images.

MAJOR:

  1. Materials and Methods section:
  2. Why did the authors decide to treat the endothelial cells with Atorvastatin after (and not before) irradiation? If Atorvastatin should be prescribed to patients in order to alleviate enteropathies following radiation, wouldn’t it be prescribed prior to the procedure? In the description of the results in the section 2.1, the authors say that “we irradiated Human umbilical vein endothelial cells (HUVECs) in the presence or absence of atorvastatin and analyzed the outcomes.” However, this is not what it is described in the Materials and Methods section. Please elaborate.
  3. How/why did the authors pick a 6h treatment period with Atorvastatin in order to perform the tube formation assay? The same question is posed for the time picked for other experiments: 24h of Atorvastatin treatment for the transendothelial migration of monocytes and immunofluorescence and 36h for the production of conditioned media. Please elaborate.
  4. In the section 4.4, please define “total segment length” and “master junction”.
  5. In the section 4.10., the authors say that “the green-positive cells per five fields were quantified and plotted”. However, no details regarding the microscope objective (e.g., magnification) were given and the readers don’t have an idea how representative five fields are in this context. Please provide more details regarding the cell counting/ microscopy including the number of cells counted per total amount of cells plated.

Results section:

1. In Fig.1B, the second bar in the bar graph refers to the migration of    endothelial cells in the presence of Atorvastatin and in the absence of irradiation. Why do the authors think that Atorvastatin induce more cell invasion in this case?

  1. In Fig.1C, especially in the Atorvastatin panel, it is very difficult to see the cells (unlabeled) due to the lack of contrast. It would be beneficial if the authors could provide images with a better contrast with respect to the background.
  2. In Fig.2C the authors measure, via WB, the amount of secreted PAI-1. However, there is no standard for comparison. It would be beneficial to include the Bradford data here, otherwise, it is really hard to interpret the data.
  3. The captions of Fig. 2E and 2F are switched. Please invert the captions or the figure panels.
  4. Please quantify the IF data represented in Fig. 3C. Otherwise it is difficult to see how representative the p-c-jun labeling is in the IR case in comparison to the other experimental groups.
  5. A very important control is missing in Fig.4: a case in which the cells are only treated with anisomycin.
  6. Are the three upper panels in Fig.4C (-Ani) the same as in Fig. 3C? Were these experiments performed on the same day and under the same conditions? Again, a quantification is needed in Fig.4C for the same reasons pointed out in the item (e).